# Storage Stability of Plant-Based Drinks Related to Proteolysis and Generation of Free Amino Acids

**DOI:** 10.3390/foods13030367

**Published:** 2024-01-23

**Authors:** Ida Schwartz Roland, Thao T. Le, Tony Chen, Miguel Aguilera-Toro, Søren Drud-Heydary Nielsen, Lotte Bach Larsen, Nina Aagaard Poulsen

**Affiliations:** 1Department of Food Science, Aarhus University, Agro Food Park 48, DK-8200 Aarhus, Denmark; 2Department of Food Science and Microbiology, Auckland University of Technology, Auckland 1010, New Zealand

**Keywords:** plant-based drinks, storage, free amino acids, proteolysis, color, shelf-life

## Abstract

The market for plant-based drinks (PBDs) is experiencing a surge in consumer demand, especially in Western societies. PBDs are a highly processed food product, and little is known about this relatively new food product category when compared to bovine milk. In the present study, the storage stability, proteolysis and generation of free amino acids were investigated in commercially available PBDs over the course of a one-year storage period. Generally, pH, color and protein solubility were found to be stable in the PBDs during storage, except for the pea-based product, which showed less protein solubility after storage. The pea-based drinks also had higher initial levels of free N-terminals prior to storage compared with levels for the other plant-based drinks, as well as significantly increasing levels of total free, and especially bitter free, amino acids. The development of free amino acids in the oat-based drink indicated that the released amino acids could be involved in various reactions such as the Maillard reaction during the storage period.

## 1. Introduction

Over the past decade, there has been a growing interest in and consumption of plant-based drinks (PBDs), driven by factors, such as lactose intolerance and allergy towards bovine milk, ethical reasons including animal welfare as well as environmental reasons such as lower carbon footprint connected to PBDs compared to bovine milk consumption [1,2,3,4]. PBDs refer to beverages made from various plant sources milled with water that resemble a milk-like texture and appearance, and they are therefore often marketed as milk replacers [5]. From a nutritional point of view, PBDs lack quality compared to bovine milk in terms of protein content and levels of essential amino acids (AAs) as well as vitamins and minerals [5,6]. However, some PBDs hold other health-promoting properties such as containing dietary fibers and being low in cholesterol [2,7] and, therefore, replacing one with another can be difficult [6].

The most predominant plant sources for PBDs are oat, almond, soy and rice, with oat-based products being the most predominant variant in the Danish market based on the number of commercial products available [8]. PBDs are highly processed food products and the processing steps usually include milling, filtration, homogenization and heat treatment [9]. Ultra-high-temperature (UHT) treatments are usually a part of the processing of PBDs to ensure food safety, but this severe treatment can produce unwanted changes in the food matrix [10]. These changes include molecular processing modifications on the present proteins and carbohydrates [8] and could be expected to develop further during storage periods, as seen in bovine UHT milk [11]. The processing-induced changes in UHT milk are known to influence nutritional quality, lead to the development of off-flavors and color changes, e.g., from the Maillard reaction, as well as potentially leading to sedimentation or age gelation [12,13]. Age gelation happens under the presence of heat-stable proteases that hydrolyze the milk proteins during storage, ultimately leading to the gelation of milk [13]. This mechanism is related to the unique properties of milk proteins, but proteolysis in PBDs might also occur during long-term storage, especially when products are stored outside the cold chain [12]. Proteolysis and the generation of free AAs could potentially also occur in PBDs and thereby influence the taste and odor of the products. This could happen both via their properties as free AAs i.e., sweet or bitter taste [14], and as part of Maillard reactions involving free AAs as observed in milk systems [11]. Free AAs as well as intact proteins can react with reducing sugars during the initial stage of the Maillard reaction, and they can play a role in the advanced stage forming Strecker aldehydes, leading to unwanted properties in the product [11]. These parameters, which can be particularly affected by the UHT treatment, i.e., gelation, proteolysis, color and off-flavor development, have not been studied as extensively in plant-based systems. Therefore, the aim of this paper was to investigate the storage stability in commercial PBDs in terms of pH and particle size, as well as color development, protein solubility, proteolysis and free AA content over a storage period of one year. We expected to observe an increase in the level of free AAs as well as in browning through color changes in PBDs that could be correlated with proteolysis and the Maillard reaction. For this study, commercially available PBDs on the Danish market were used.

## 2. Materials and Methods

### 2.1. Samples of Plant-Based Drinks and Experimental Set-Up

Seven different commercially available PBDs were selected for this study. The plant sources of the drinks comprised one oat, one oat/hemp in combination, two almond drinks, one pea, one soy and one soy/rice combination (Table 1). The drinks were selected to represent the Danish market of PBDs, as well as a variety of plant sources for PBDs. Declared nutritional contents and shelf-life of the studied drinks are provided in Table 1, and product names and manufacturers are listed in Appendix A. All drinks were produced within 1–3 weeks prior to the start of the experiment.

For investigation of storage stability and potential storage-induced changes, the PBDs were stored at 21 °C for 364 days, comprising 11 sampling time points on days 0, 14, 28, 49, 91, 133, 175, 217, 259, 301 and 364. Fresh sample analyses conducted on each day of sampling comprised determination of color and pH. Samples were then aliquoted, frozen at −80 °C and then transferred to −20 °C and stored until further analysis for other parameters.

### 2.2. Color and pH Measurement

Color development of stored samples of PBDs was monitored using a Chroma-meter (CR-400, Konica Minolta Inc., Tokyo, Japan) on 10 mL samples of PBDs held in a ceramic crucible. Before the measurements, calibration of the chroma-meter was conducted with a standard plate, and the measurements were taken in technical triplicates. The CIELAB system was used as the color space application [15], where L* is the black–white scale, a* is the red–green scale and b* is the yellow–blue scale. Hence, L* = 0 is black and L* = 100 is white; negative a* is green and positive a* is red; and negative b* is blue and positive b* is yellow. Data for days 0, 175 and 364 are shown in the results section, and the remaining data points are available in Appendix A. The pH was measured using a pH meter (PHM 92, Radiometer, Copenhagen, Denmark) calibrated with pH standards (Hanna Instruments, Woonsocket, RI, USA). Data for days 0, 175 and 364 are shown in the results section, and the rest of the data points are available in Appendix A.

### 2.3. Protein Content and Protein Stability during Storage

Protein contents were stated on the package, Table 1. Furthermore, total nitrogen content was assessed through Dumas analysis, employing Dumatherm (Gerhardt, Königswinter, Germany). Values of total N were measured in the PBDs on day 0, as well as days 0 and 364 of storage for PBD supernatants after centrifugation for 10 min at 10,000× *g* at 4 °C to investigate protein solubility before and after storage. Via comparison of protein content on day 0 before and after centrifugation and after centrifugation at day 364, protein solubility was investigated. The samples were measured in technical duplicates using 0.5% tris(hydroxymethyl)aminomethane as a standard. For each plant source, the applied nitrogen to protein conversion factor was oat = 5.83, almond = 5.18, pea = 5.3, soy = 5.71 and hemp seeds = 5.3 [16].

### 2.4. Particle Size Distribution

Development in particle size distribution in the PBDs was determined using Zetasizer Lab (Malvern Panalytical, Malvern, UK) [8]. The PBD samples were diluted at a ratio of 1:50 in Milli-Q water and assessed using a 1 × 1 cm cuvette. Milli-Q water (refractive index of 1.33) was in the continuous phase while PBD constituted the scattered phase. Measurement and analysis were conducted using the Zetasizer Lab instrument and processed through the ZS Xplorer software (version 2.0.0.98, Malvern Panalytical, Malvern, UK).

### 2.5. Free N-Terminals via o-Phthalaldehyde Assay

Development of free N-terminals was measured with the spectrophotometric method o-phthalaldehyde (OPA) assay. Samples of the stored PBDs were precipitated on ice for 30 min with 5% trichloroacetic acid and centrifuged at 13,000× *g* for 20 min at 4 °C. A 10 µL supernatant was mixed with 200 µL OPA reagent (0.1 M Na_2_B_4_O_7_, 0.1% SDS, 5.7 mM DTE, 1% OPA) in triplicate in a 96-well microtiter plate and incubated for 15 min at 21 °C. The absorbance was measured at 340 nm in a plate reader (BioTek Synergy 2, Holm & Halby, Brøndby, Denmark) and the standard curve was glutamine (1.25–40 mM).

### 2.6. Quantification of Free Amino Acids on Days 0, 175 and 364 of Storage Period Using Triple-Quadrupole Mass Spectrometry

Ten mg freeze-dried PBD was weighed off and precipitated on ice with 250 µL of 50% *v*/*v* methanol and centrifuged at 10,000× *g* for 10 min at 4 °C to extract free AAs. The supernatant was collected, and the pellet was resuspended in 250 µL 80% *v*/*v* methanol and precipitated on ice, then being centrifuged a second time at 10,000× *g* for 10 min at 4 °C. The two supernatants were combined and 40 µL was mixed with an equal volume of methanol containing 10 mg/L d4-alanine internal standard. The mixture was vortexed and centrifuged at 10,000× *g* for 5 min at 4 °C. The supernatant was derivatised based on the AccuTag method as previously described [17,18]. The AccuTag reagent was 6-aminoquinolyl-N-hydroxysuccinimidyl carbamate, 2.8 mg in 1 mL dry acetonitrile, and the derivatization reaction occurred in 70 µL borate buffer with 10 µL sample, standard or blank and 10 µL Accutag reagent. The reaction was incubated for 15 min at 55 °C followed by addition of 400 µL 90% formic acid, and the samples were ready for analysis. The standard solution of AA was A9906 (Sigma-Aldrich, Darmstadt, Germany).

The liquid chromatography–mass spectrometry (LC-MS) analysis method was conducted on an Agilent 1260 Infinity Quaternary LC system (Santa Clara, CA, USA) coupled to a 6420 Triple-Quadrupole Mass Spectrometer (triple Q-MS) with an electrospray ionization source. The LC-MS was equipped with a Kinetex Evo C18 2.1 × 150 mm, 1.7 µm (Phenomenex, Torrance, CA, USA) and ran at 25 °C with a flow rate of 0.25 mL/min. The mobile phases were 0.1% formic acid in Milli-Q water (A) and 0.1% formic acid in acetonitrile (B), and the gradient program was set as follows: 0–8 min: 95% A, 8–13 min: 90% A, 13–16.5 min: 85% A, 16.5–18.5: 20% A, 18.5–29 min: 95% A. The MS ionisation source conditions were as follows: capillary voltage of 4 kV, drying gas temperature of 300 °C, drying gas flow of 10 L/min, nebulizer pressure of 30 psi. Positive ion mode was utilized with multiple reaction monitoring (MRM) for quantitative analysis. Data analysis was conducted in MassHunter Qualitative Analysis version 10.0 and MassHunter Quantitative Analysis version 10.1 for QQQ (Agilent Technologies, Santa Clara, CA, USA). Results for proteinogenic AAs are shown in the results section whereas results for non-proteinogenic AAs are available in Appendix A. 

### 2.7. Statistical Analysis

Statistical analysis was conducted using the RStudio software (version 2023.6.1., Boston, MA, USA) and Microsoft Excel (2016). Significant differences were tested for Z-average particle sizes over time with one-way analysis of variance (ANOVA) in Excel as well as one-tail *t*-test for analysing the protein solubility. Variances in the free amino acid content were evaluated through mean comparisons utilizing Tukey’s HSD post hoc test subsequent to conducting a one-way ANOVA test the data. Statistical significance was established at *p* ≤ 0.05.

## 3. Results

### 3.1. Color, pH and Protein Solubility

pH and color were measured at each sampling time and are summarized in Table 2 for sampling days 0, 175 and 364. Data for all 11 sampling days can be seen in the Appendix A. The off-set pH for the samples ranged from 6.77 (oat) to 8.08 (almond), while the end pH after 364 days of storage ranged from 6.39 (oat/hemp) to 7.61 (pea). The stored samples slightly decreased in pH over the storage period except for pea and soy PBDs, where the pH did not change during storage. For the color measurements, the oat/hemp showed the lowest values for L* (black and white scale), meaning that it was the darkest PBD at the end of the storage period. The rest of the samples had L*-values between 52 and 64, and all samples decreased in whiteness over time. The soy sample was the most green and yellow according to the a* and b* values, whereas the almond samples were the least yellow. All samples showed decreasing values of b* (yellowness) over the storage period, whereas parameter a* (red-green scale) changed for oat/hemp and the two soy drinks over time.

Protein content was measured based on total N in the original PBDs, and the contents ranged from 0.44% (roasted almond) to 4.22% (soy) (Table 2). Protein solubility and thereby indications of sedimentation involving protein were measured as an expression of the protein remaining in the supernatant after centrifugation, both on day 0 and day 364. Apart from the soy/rice, the PBDs had significantly higher protein content measured as total N in the original PBD, as compared with the supernatant after centrifugation on day 0 prior to storage (Table 2), indicating the presence of proteinaceous sediment. The soy/rice PBD was the only sample containing the same levels of protein in the intact sample, as in the centrifuged samples on day 0, thereby indicating no sedimentation involving protein. Comparing total protein contents in the supernatants of the PBDs, day 0 and day 364 after storage showed a statistically different increase for pea, while increases for roasted almond and soy were not statistically different. This showed that there was no decrease in protein solubility over time but rather, that in some PBDs it seemed that the protein solubility increased with storage, especially for pea. For the soy/rice, there was no difference in total protein contents in supernatants from day 0 to day 364 of storage, indicating a very stable protein matrix in relation to solubility.

### 3.2. Particle Size

The development in particle size for each PBD during the storage period was measured and expressed as a Z-average (Figure 1). The Z-average particle size of the samples ranged from 1633 nm (almond) to 393 nm (oat), exhibiting large heterogeneity between the samples. The final Z-average particle sizes showed less heterogeneity, with the largest size being 897 nm (oat hemp) and the smallest particle size being 399 nm (soy). All PBDs changed significantly over time (*p* ≤ 0.05), with oat and pea increasing in Z-average over time, especially after 259 days of storage, whereas the other five PBDs decreased during the storage period. Almond rapidly decreased within the first 28 days, whereas oat/hemp increased in the same period and generally displayed a large fluctuation in particle size. All Z-average particle sizes changed significantly over time (*p* ≤ 0.05); oat and pea PBDs displayed an increase in particle size during storage, whereas oat/hemp, roasted almond, almond, soy and soy/rice more or less clearly decreased in particle size.

### 3.3. Proteolysis and Generation of Free Amino Acids during Storage

Protein degradation during storage as a result of potential proteolysis was determined as the level of free N-terminals using the OPA assay (Figure 2) and free AA content (Table 3). Almond and roasted almond had the lowest concentrations of free N-terminals, with average values < 1 µM on day 0 and for the whole storage period. The pea sample had the highest levels of free AAs, with an initial concentration of 18.1 µM and a final concentration of 23.8 µM (cf., right *y*-axis). The oat PBD and soy PBD were similar in free N-terminal content and remained constant over the storage period, except for the oat PBD, which increased after 259 days of storage.

To investigate the specific AAs that may be released over time, the levels of free AA were quantified in the stored PBD samples on sampling days 0, 175 and 364 (Table 3) using quantitative triple Q MS-based analysis. The summarized data show the sum of all free AAs in the samples. The initial contents ranged from 1.2 to 35.6 mg free AA/g protein, with min. and max. representing soy and oat, respectively, whereas the final contents ranged from 2.5 to 29.2 mg free AA/g protein, with min. and max. representing soy and oat, respectively. Total free AA in oat PBD was significantly higher on day 175 compared to levels on both days 0 and 364, whereas for pea PBD, free AA levels were significantly higher on days 175 and 364 compared to day 0. The remaining PBDs did not differ in the content of total free AAs across the three time points. For almond and roasted almond, this pattern was also reflected for individual AAs, as these PBDs did not differ between any time points among all quantified AAs. The soy drinks also showed the same levels of free AAs at all time points, except for tryptophan, which significantly increased on day 364. Oat PBD showed significantly different levels of free AAs for most of the quantified AAs over the storage period. L-alanine, cystine and glycine significantly increased over the storage time in oat, whereas aspartic acid, asparagine, threonine and valine decreased between days 175 and 364; glutamic acid, proline, tryptophan and tyrosine decreased over the whole storage period. In the pea drink, L-alanine, lysine, methionine, serine, threonine and valine increased significantly at all three time points, whereas glycine, isoleucine, leucine, phenylalanine, tyrosine and arginine significantly increased between days 0 and 175 but not thereafter. For the oat/hemp drink, the contents of free lysine and methionine were significantly increased on day 364 compared to day 0, and the content of free glutamine was significantly decreased on day 364 compared to day 0, whereas for all other free AAs in oat/hemp, the contents did not significantly differ.

## 4. Discussion

This paper aimed to examine the storage stability of commercially available PBDs from the Danish market, concerning the proteolysis and generation of free amino acids. Therefore, parameters related to protein solubility and proteolysis, including the level and composition of free AAs, color and particle size, were measured in different types of PBDs stored for one year. Previous studies found that particularly the UHT treatment of bovine milk resulted in reduced storage stability due to heat- and processing-induced reactions, like the Maillard reaction and/or dehydroalanine pathway and proteolysis, potentially contributing to age gelation [12,13]. After UHT treatment, both milk and PBD are sterile and aseptically packed; however, some enzymes are heat-resistant and may withstand the UHT treatment [13]. In bovine milk, some indigenous enzymes are found to be heat-stable, e.g., those from the plasmin system, but enzymes from psychotropic bacteria can also be found in UHT milk from contamination of the raw milk [12]; hence, it has been investigated if these mechanisms could be present and quantifiable in PBDs.

As outlined in Table 1, PBDs have some inherent differences e.g., in protein contents, but these did not seem to be primary drivers for the subsequent differences in the solubility and sedimentation of protein. All PBDs (except the soy/rice PBD), representing big variations in protein contents, displayed the presence of sedimentable proteinaceous material prior to storage. The soy/rice PBD, containing the second-highest content of total protein among all PBDs studied here, was the most stable in relation to protein solubility, both before and after storage. Perhaps surprisingly, no significant decreases in total protein contents in the supernatants before and after storage were observed for any PBDs, indicating that there was no significant protein sedimentation during storage. In contrast, there were indications of increased protein solubility with time, especially for pea. This is known to be a problem in some PBDs, in that sediment is observed to develop and may be considered a quality problem for consumers [19]. Soy PBDs contained the highest protein concentrations and almond PBDs the lowest, but according to the measurements of total protein after centrifugation on day 0 and day 364, these drinks seemed stable during storage, as the total protein contents of supernatants before and after storage were not significantly different. The results here indicate that any sediment developing may be more related to non-proteinaceous material like fibres.

The particle size was influenced over time i.e., oat and pea PBD had increasing particle size towards the end of the experiment, whereas the other PBDs showed decreasing values. A previous study concluded that smaller particle sizes were associated with more stable PBDs, and larger particle sizes produced less stable drinks [20]. This relationship was not observed here using protein solubility as a parameter for stability, i.e., for pea, the particle size increased over time, but the protein solubility also increased. It is likely that in PBDs, there is no direct correlation between protein solubility and particle size.

Heat treatment through e.g., roasting of the starting material, may affect the color of PBDs, which is also apparent from Table 2, comparing almond and roasted almond, where the b* values of roasted almond are much higher compared to almond. A previous study quantified compounds related to the Maillard reaction in the same two almond drinks and found no significant difference in the levels of those compounds in non-stored PBDs; however, it is still necessary to quantify the compounds in stored PBDs [8].

Looking at parameters for proteolysis, based on the OPA assay, the level of free N-terminals in the pea PBD increased continuously over time, whereas the oat PBD increased rapidly towards the end of the storage period, and the other drinks remained relatively constant (Figure 2). For oat, there seemed to be an accelerated effect between days 175 and 364, as the level of free N-terminals was increased on days 301 and 364 compared to the remaining storage period. However, this was not reflected in total free AAs, which were highest on day 175 compared to lower total free AAs on day 364. The determination of free N-terminals via the OPA assay comprises levels of free N-terminals in intact plant proteins, generated peptides and free AAs, as well as lysine side chains. Changes in lysine side chains in proteins and peptides could also potentially contribute to the changes observed, i.e., like engaging in Maillard or dehydroalanine pathway reactions [11,12]. The level measured by the OPA assay is therefore, a balance between any proteolysis occurring, creating peptides and free AAs on the side, and then the stability of both proteins and peptides in relation to engaging in Maillard or dehydroalanine pathway reactions, as well as the generated free AAs engaging in, e.g., Maillard reactions via Strecker aldehydes [21]. This may also explain why lower levels of free AAs were seen in the oat PBD at the same time as higher levels of free N-terminals [21]. Furthermore, it is known from the processing of oat drinks that the oat slurry needs to be hydrolysed by alpha-amylases to prevent gelatinization after cooling down [22]. The utilized alpha-amylase needs to be both heat-stable in order to function efficiently during the processing of the PBDs and of high purity to avoid the presence of proteolytic side activity.

High and increasing levels of free N-terminals in pea PBD, as determined by the OPA assay, were also reflected in high and increasing total free AAs, as determined by triple Q-MS. Levels of free AAs in pea significantly increased from days 0 to 175, which, for pea, confirmed proteolysis and the generation of free AAs occurred during storage. What makes the pea PBD more prone to proteolysis is unclear, but the initial high level of amino N-terminals suggests that it is related to the raw material, which, in this case, is pea protein, as listed in Appendix A. During protein fractionation, proteolysis may occur since proteases and other enzymes are released in the initial steps of the purification process, which could be the reason for the high initial levels of free N-terminals in pea PBD [23]. This indicates some kind of proteolytic activity being present in the pea PBD, however, the identity of such protease is not known at present. It can be speculated whether this increase in proteolysis in the pea PBD could be the driver behind the increased solubility of total pea protein observed in the pea PBD during storage, and if the formed peptides of free AAs are more soluble than intact protein.

A previous study concluded that the formation of free AAs in UHT milk contributed to stale aromas, which could be linked to the formation of Strecker aldehydes [24]. Another study classified free AAs in food systems into groups based on how they impact flavour and found that ten AAs contribute to bitterness, i.e., arginine, histidine, tyrosine, leucine, valine, methionine, isoleucine, phenylalanine, lysine and proline [14]. In this study, we found nine out of the ten bitter free AAs significantly increased during the storage period in pea PBD, and, therefore, conducting sensory analysis on these stored PBDs could be relevant. Free serine, alanine, arginine, glycine and threonine contribute to sweet aromas [14,25], and in pea PBD, all five of these were found to have significantly higher levels after the storage period compared to the initial level; oat alanine and glycine were observed to increase significantly during storage.

Total free AA was standardized to protein content, but it would be interesting to determine the total AA compositions of proteins in the PBDs in order to better interpret differences across, e.g., oat and pea. It is, however, clear that the free AAs in the two PBDs are dominated by different free AAs, and this could affect the nutritional value of the drinks differently and probably relates to the total AA profile of the protein sources. Pea is for instance high in branched-chain AAs, like valine, leucine and isoleucine [26], and the concentrations of these free AAs are all generally high in pea PBD and increasing during storage. The underlying enzymatic or non-enzymatic drivers still need to be confirmed. Table 3 shows the level of total free AA in 100 g of PBD, i.e., the values here are not standardized to protein content but represent the absolute level, as experienced in the drinks. From these values, it is evident that pea contains the highest content of free AAs and therefore, could be most affected by their presence since the free AA will be available for further reactions, like the formation of Strecker aldehydes as part of the Maillard reaction [11].

## 5. Conclusions

In the present study, the possible quality deteriorations of commercially available PBDs were studied over the course of a one-year storage period, especially in relation to color and pH, particle size, proteolysis and the generation of free AAs. It was found that the parameters varied depending on plant sources. For particle size, the oat/hemp PBD had the highest particle size. Limited variations were seen in pH, color and protein solubility over time, whereas protein solubility significantly increased in the pea PBD. Oat had the highest total level of free AAs relative to protein content, whereas pea had the highest content of free AAs relative to g PBD. The pea PBD was found to have a significantly increased amount of free AAs over the storage period, in line with pea PBD also having the highest increase in the level of total free N-terminals over storage, indicating that proteolysis could have occurred. Oat significantly increased from days 0 to 175 and decreased to the initial levels again on day 364. In conclusion, it is clear that the plant source starting material is important for the specific changes occurring during storage in PBDs. No gelation was found, but the observed changes in free AAs may affect the flavour of the products. Generally, the underlying molecular mechanisms for the observed changes should be the focus of further studies, where the changes in amino acid content could be elucidated by specific reactions, such as the Maillard reaction and protein crosslinking through the dehydroalanine pathway, and the effect of digestibility may be highly relevant.

## Figures and Tables

**Figure 1 foods-13-00367-f001:**
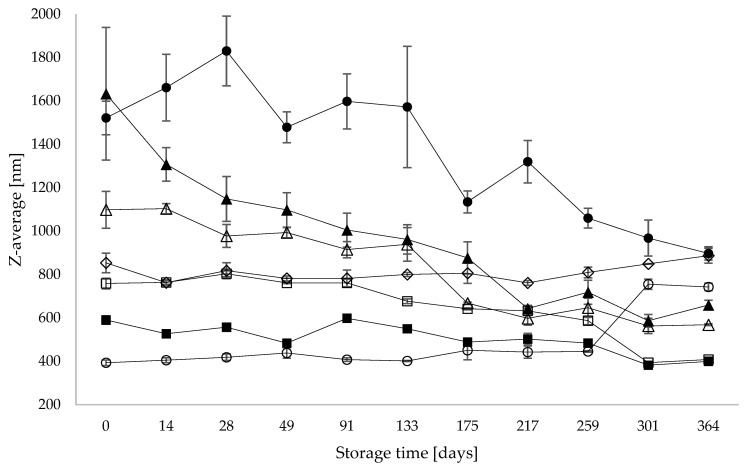
Particle size measured as Z-average (nm) on stored PBD. Oat (○), oat/hemp (●), roasted almond (Δ), almond (▲), soy (□), soy/rice (■) and pea (◊). All samples were measured in technical triplicates (n = 3).

**Figure 2 foods-13-00367-f002:**
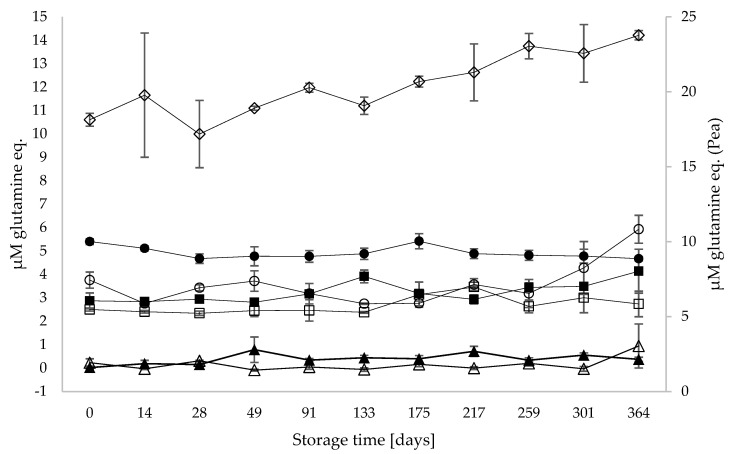
Level of free N-terminals in stored PBD expressed as glutamine equivalents (µM). Right *y*-axis refers to PBD pea (◊), whereas left *y*-axis refers to all other PBDs, oat (○), oat/hemp (●), roasted almond (Δ), almond (▲), soy (□), and soy/rice (■). Results represent technical triplicates (n = 3).

**Table 1 foods-13-00367-t001:** The nutritional composition of the studied PBDs, as stated on their packaging.

	Oat	Oat/Hemp	Almond	Roasted Almond	Pea	Soy	Soy/Rice
Energy (kJ/kcal)	219/52	205/48	87/21	89/21	142/34	147/35	231/55
Protein (g/100 g)	0.4	1.1	0.5	0.4	2	3.7	3.5
Fat (g/100 g)	1.9	1.0	1.2	0.9	2.1	2.1	2.0
Saturated fat (g/100 g)	0.3	0	0.1	0.1	0.2	0.4	0.3
Carbohydrates (g/100 g)	8	8.7	2	2.7	2	0.6	5.6
Sugars (g/100 g)	3.2	3.3	1.9	2.4	2	0.1	4.0
Fibre (g/100 g)	1	N/A	0.1	0.3	N/A	0.6	0.5
Salt (g/100 g)	0.08	0.1	0.1	0.1	0.2	0.04	0.19
Calcium (mg/100 g)	N/A	N/A	120	120	120	N/A	120
Shelf life ^1^	9 mo., RT	1–2 mo., 5 °C	9 mo., RT	9 mo., RT	12 mo., RT	9 mo., RT	9 mo., RT
Heat treatment	UHT	ESL	UHT	UHT	UHT	UHT	UHT

Room temperature (RT). Ultra-high temperature (UHT). Extended shelf life (ESL). ^1^ From manufacturers.

**Table 2 foods-13-00367-t002:** pH, color parameters (L*, a*, b*) and protein solubility measurement of stored PBDs on sampling days 0, 175 and 364. For pH and color parameters, the standard deviations were all under 3% and not included for better readability. Different subscript letters for protein solubility for each PBD indicate statistical significance, (*p* ≤ 0.05).

Sample	Day	pH	L*	a*	b*	Protein PBD (%)	Protein Supernatant (%)
Oat	0	6.77	58.89	−0.33	10.02	0.456 ± 0.007 ^b^	0.164 ± 0.018 ^a^
	175	6.45	54.41	0.20	9.93		
	364	6.54	53.97	0.44	9.92		0.226 ± 0.001 ^a^
Oat/hemp	0	6.80	54.42	−1.51	8.46	0.930 ± 0.003 ^b^	0.328 ± 0.000 ^a^
	175	6.32	49.55	−0.67	7.60		
	364	6.39	48.55	−0.40	7.47		0.334 ± 0.002 ^a^
Roasted Almond	0	7.60	55.84	0.48	7.87	0.441 ± 0.005 ^b^	0.265 ± 0.000 ^a^
	175	7.08	52.32	0.65	7.18		
	364	7.37	52.43	0.44	6.93		0.368 ± 0.002 ^ab^
Almond	0	8.08	59.32	−0.19	2.84	0.449 ± 0.019 ^b^	0.310 ± 0.003 ^a^
	175	7.42	55.61	−0.01	2.26		
	364	7.50	55.60	−0.19	2.25		0.312 ± 0.125 ^a^
Pea	0	7.62	63.75	−1.40	8.41	1.702 ± 0.035 ^c^	1.144 ± 0.003 ^a^
	175	7.45	59.74	−1.02	8.28		
	364	7.61	59.57	−1.03	8.01		1.173 ± 0.000 ^b^
Soy	0	7.18	62.48	−3.00	11.57	4.223 ± 0.182 ^b^	3.201 ± 0.033 ^a^
	175	7.05	58.71	−2.49	10.27		
	364	7.24	58.40	−2.16	9.42		3.222 ± 0.001 ^ab^
Soy/rice	0	7.94	58.70	−1.10	10.73	2.617 ± 0.584 ^a^	2.884 ± 0.065 ^a^
	175	7.56	53.37	0.16	9.80		
	364	7.54	52.85	0.52	9.68		2.984 ± 0.000 ^a^

**Table 3 foods-13-00367-t003:** Contents of free AAs in PBD at days 0, 175 and 364 in mg AA/g protein (n = 3). Letters indicate significant difference (*p* ≤ 0.05) in contents of each free AA in a PBD over the storage period, for better readability, letters were not included for free AAs with no significant change (*p* > 0.05). Not detected (ND).

Free AA	Day	Oat	Oat/Hemp	Almond	Roasted Almond	Pea	Soy	Soy/Rice
Histidine	0	0.18 ± 0.00	0.05 ± 0.00	ND	ND	0.22 ± 0.05 ^a^	0.04 ± 0.00	0.17 ± 0.01
	175	0.27 ± 0.05	0.06 ± 0.00	ND	ND	0.37 ± 0.07 ^ab^	0.07 ± 0.01	0.23 ± 0.05
	364	0.15 ± 0.00	0.05 ± 0.00	ND	ND	0.48 ± 0.11 ^b^	0.14 ± 0.00	0.16 ± 0.03
Isoleucine	0	0.25 ± 0.01	0.13 ± 0.01	0.10 ± 0.01	0.09 ± 0.00	0.64 ± 0.04 ^a^	0.02 ± 0.00	0.05 ± 0.00
	175	0.27 ± 0.03	0.13 ± 0.01	0.09 ± 0.01	0.09 ± 0.00	1.14 ± 0.20 ^b^	0.02 ± 0.00	0.05 ± 0.00
	364	0.19 ± 0.01	0.17 ± 0.02	0.10 ± 0.02	0.10 ± 0.01	1.30 ± 0.09 ^b^	0.04 ± 0.00	0.05 ± 0.00
Leucine	0	0.21 ± 0.00	0.29 ± 0.03	0.05 ± 0.01	0.03 ± 0.00	2.20 ± 0.13 ^a^	0.01 ± 0.00	0.06 ± 0.00
	175	0.24 ± 0.03	0.32 ± 0.03	0.06 ± 0.01	0.02 ± 0.00	3.60 ± 0.57 ^b^	0.02 ± 0.00	0.06 ± 0.00
	364	0.09 ± 0.00	0.38 ± 0.04	0.06 ± 0.01	0.03 ± 0.00	4.00 ± 0.17 ^b^	0.07 ± 0.00	0.06 ± 0.01
Lysine	0	0.13 ± 0.03	0.60 ± 0.09 ^a^	0.28 ± 0.04	0.05 ± 0.00	1.42 ± 0.05 ^a^	0.06 ± 0.01	0.05 ± 0.00
	175	0.34 ± 0.04	0.84 ± 0.10 ^ab^	0.30 ± 0.06	0.12 ± 0.03	3.24 ± 0.35 ^b^	0.08 ± 0.02	0.07 ± 0.00
	364	0.61 ± 0.02	1.19 ± 0.03 ^b^	0.31 ± 0.05	0.19 ± 0.01	3.77 ± 0.35 ^c^	0.04 ± 0.00	0.05 ± 0.01
Methionine	0	0.02 ± 0.00	0.05 ± 0.00 ^a^	0.02 ± 0.00	0.01 ± 0.00	0.08 ± 0.01 ^a^	0.01 ± 0.00	0.02 ± 0.00
	175	0.02 ± 0.00	0.08 ± 0.01 ^b^	0.02 ± 0.00	0.02 ± 0.00	0.14 ± 0.01 ^b^	0.01 ± 0.00	0.03 ± 0.00
	364	0.01 ± 0.00	0.08 ± 0.01 ^b^	0.01 ± 0.00	0.01 ± 0.00	0.18 ± 0.01 ^c^	0.02 ± 0.00	0.02 ± 0.00
Phenylalanine	0	0.16 ± 0.01	0.14 ± 0.01	0.07 ± 0.01	0.01 ± 0.01	0.76 ± 0.07 ^a^	0.01 ± 0.00	0.04 ± 0.00
	175	0.24 ± 0.01	0.15 ± 0.02	0.07 ± 0.02	0.01 ± 0.00	1.45 ± 0.27 ^b^	0.01 ± 0.00	0.05 ± 0.01
	364	0.24 ± 0.00	0.19 ± 0.03	0.07 ± 0.01	0.02 ± 0.00	1.54 ± 0.12 ^b^	0.05 ± 0.01	0.04 ± 0.00
Threonine	0	0.40 ± 0.03 ^ab^	0.17 ± 0.01	0.09 ± 0.02	0.06 ± 0.01	0.35 ± 0.07 ^a^	0.01 ± 0.00	0.02 ± 0.00
	175	0.50 ± 0.08 ^b^	0.19 ± 0.03	0.09 ± 0.01	0.04 ± 0.01	0.65 ± 0.09 ^b^	0.01 ± 0.00	0.03 ± 0.00
	364	0.27 ± 0.01 ^a^	0.25 ± 0.03	0.08 ± 0.01	0.07 ± 0.01	0.82 ± 0.04 ^c^	0.01 ± 0.00	0.02 ± 0.00
Tryptophan	0	0.95 ± 0.02 ^b^	0.11 ± 0.02	0.05 ± 0.01	0.03 ± 0.00	0.02 ± 0.00	0.10 ± 0.02 ^a^	0.24 ± 0.01
	175	1.16 ± 0.20 ^c^	0.11 ± 0.01	0.04 ± 0.01	0.03 ± 0.01	0.02 ± 0.00	0.11 ± 0.02 ^a^	0.24 ± 0.02
	364	0.30 ± 0.04 ^a^	0.15 ± 0.02	0.04 ± 0.01	0.02 ± 0.00	0.04 ± 0.01	0.35 ± 0.02 ^b^	0.22 ± 0.01
Valine	0	0.45 ± 0.01 ^ab^	0.24 ± 0.02	0.15 ± 0.01	0.08 ± 0.00	0.61 ± 0.05 ^a^	0.02 ± 0.00	0.05 ± 0.00
	175	0.59 ± 0.12 ^b^	0.26 ± 0.03	0.13 ± 0.02	0.08 ± 0.01	1.10 ± 0.02 ^b^	0.02 ± 0.00	0.06 ± 0.00
	364	0.33 ± 0.01 ^a^	0.36 ± 0.03	0.15 ± 0.04	0.11 ± 0.03	1.43 ± 0.08 ^c^	0.04 ± 0.00	0.05 ± 0.01
L-Alanine	0	1.46 ± 0.03 ^a^	0.90 ± 0.06	0.70 ± 0.09	0.72 ± 0.10	0.79 ± 0.04 ^a^	0.11 ± 0.01	0.22 ± 0.01
	175	1.88 ± 0.33 ^a^	1.07 ± 0.09	0.79 ± 0.09	0.64 ± 0.06	1.97 ± 0.16 ^b^	0.15 ± 0.02	0.24 ± 0.01
	364	2.45 ± 0.09 ^b^	1.28 ± 0.08	0.67 ± 0.09	0.79 ± 0.07	2.50 ± 0.12 ^c^	0.11 ± 0.01	0.22 ± 0.02
Arginine	0	ND	ND	0.24 ± 0.08	ND	0.62 ± 0.18 ^a^	0.23 ± 0.07	0.80 ± 0.16
	175	1.26 ± 0.22	0.42 ± 0.39	ND	ND	1.35 ± 0.26 ^b^	0.27 ± 0.15	1.05 ± 0.21
	364	ND	0.08 ± 0.00	ND	ND	1.45 ± 0.06 ^b^	0.63 ± 0.03	0.88 ± 0.05
Asparagine	0	12.30 ± 0.45 ^a^	1.97 ± 0.17	1.22 ± 0.13	0.72 ± 0.16	0.84 ± 0.06	0.07 ± 0.00	0.43 ± 0.01
	175	15.39 ± 2.30 ^b^	2.07 ± 0.13	1.30 ± 0.30	0.72 ± 0.04	1.79 ± 0.22	0.07 ± 0.02	0.45 ± 0.03
	364	9.49 ± 1.35 ^a^	3.02 ± 0.09	1.19 ± 0.19	0.82 ± 0.14	1.68 ± 0.20	0.12 ± 0.01	0.41 ± 0.03
Aspartic acid	0	4.14 ± 0.01 ^a^	0.73 ± 0.11	0.59 ± 0.07	0.55 ± 0.09	0.15 ± 0.03	0.15 ± 0.03	0.32 ± 0.01
	175	5.37 ± 0.86 ^b^	0.82 ± 0.05	0.66 ± 0.15	0.50 ± 0.03	0.35 ± 0.04	0.18 ± 0.04	0.35 ± 0.02
	364	3.35 ± 0.16 ^a^	1.01 ± 0.15	0.59 ± 0.08	0.68 ± 0.09	0.49 ± 0.02	0.26 ± 0.01	0.33 ± 0.02
Cystine	0	0.52 ± 0.02 ^a^	0.53 ± 0.07	0.32 ± 0.04	0.30 ± 0.07	0.33 ± 0.02 ^a^	0.03 ± 0.01	0.06 ± 0.01
	175	0.7 ± 0.16 ^a^	0.64 ± 0.07	0.37 ± 0.10	0.31 ± 0.03	1.14 ± 0.17 ^b^	0.04 ± 0.02	0.06 ± 0.00
	364	1.62 ± 0.29 ^b^	0.86 ± 0.11	0.33 ± 0.06	0.40 ± 0.07	1.44 ± 0.13 ^b^	0.02 ± 0.00	0.05 ± 0.01
Glutamic acid	0	7.00 ± 0.01 ^b^	0.89 ± 0.07	0.61 ± 0.07	0.14 ± 0.01	0.29 ± 0.01	0.16 ± 0.03	0.53 ± 0.02
	175	8.96 ± 1.43 ^c^	1.09 ± 0.09	0.66 ± 0.16	0.10 ± 0.01	0.50 ± 0.03	0.21 ± 0.04	0.57 ± 0.04
	364	3.85 ± 0.25 ^a^	1.35 ± 0.14	0.59 ± 0.06	0.08 ± 0.01	0.55 ± 0.00	0.39 ± 0.02	0.52 ± 0.06
Glutamine	0	ND	0.19 ± 0.03 ^b^	ND	ND	0.01 ± 0.00	ND	ND
	175	ND	0.12 ± 0.02 ^ab^	ND	ND	0.01 ± 0.00	ND	ND
	364	ND	0.10 ± 0.03 ^a^	ND	ND	ND	ND	ND
Glycine	0	0.52 ± 0.02 ^a^	0.53 ± 0.07	0.32 ± 0.04	0.30 ± 0.07	0.33 ± 0.02 ^a^	0.03 ± 0.01	0.06 ± 0.01
	175	0.70 ± 0.16 ^a^	0.64 ± 0.07	0.37 ± 0.10	0.31 ± 0.03	1.14 ± 0.17 ^b^	0.04 ± 0.02	0.06 ± 0.00
	364	1.62 ± 0.29 ^b^	0.86 ± 0.11	0.33 ± 0.06	0.40 ± 0.07	1.44 ± 0.13 ^b^	0.02 ± 0.00	0.05 ± 0.01
Proline	0	5.15 ± 0.15 ^b^	0.67 ± 0.02	1.05 ± 0.14	0.68 ± 0.06	0.05 ± 0.01	0.03 ± 0.00	0.07 ± 0.00
	175	6.46 ± 1.01 ^c^	0.75 ± 0.05	1.07 ± 0.19	0.61 ± 0.03	0.09 ± 0.01	0.03 ± 0.01	0.08 ± 0.00
	364	3.99 ± 0.17 ^a^	0.94 ± 0.03	1.02 ± 0.13	0.72 ± 0.04	0.15 ± 0.01	0.04 ± 0.00	0.07 ± 0.01
Serine	0	0.60 ± 0.04	0.34 ± 0.03	0.29 ± 0.00	0.26 ± 0.06	1.03 ± 0.05 ^a^	0.02 ± 0.00	0.06 ± 0.01
	175	0.77 ± 0.09	0.41 ± 0.04	0.32 ± 0.08	0.23 ± 0.01	2.89 ± 0.36 ^b^	0.03 ± 0.01	0.06 ± 0.01
	364	0.95 ± 0.07	0.58 ± 0.05	0.28 ± 0.06	0.31 ± 0.04	3.37 ± 0.38 ^c^	0.04 ± 0.01	0.06 ± 0.01
Tyrosine	0	0.44 ± 0.02 ^b^	0.09 ± 0.01	0.04 ± 0.01	0.03 ± 0.01	0.18 ± 0.03 ^a^	0.01 ± 0.00	0.05 ± 0.01
	175	0.55 ± 0.09 ^b^	0.10 ± 0.02	0.05 ± 0.00	0.03 ± 0.00	0.47 ± 0.09 ^b^	0.01 ± 0.00	0.06 ± 0.00
	364	0.29 ± 0.00 ^a^	0.14 ± 0.02	0.03 ± 0.00	0.04 ± 0.00	0.52 ± 0.02 ^b^	0.02 ± 0.00	0.05 ± 0.00
Total free AA	0	35.53 ± 0.87 ^a^	8.72 ± 0.82	6.67 ± 0.83	4.46 ± 0.68	10.67 ± 0.91 ^a^	1.22 ± 0.19	3.69 ± 0.29
	175	46.31 ± 7.26 ^b^	10.30 ± 1.24	6.83 ± 1.37	4.21 ± 0.29	22.40 ± 2.94 ^b^	1.52 ± 0.39	4.19 ± 0.44
	364	29.44 ± 2.61 ^a^	13.10 ± 1.00	6.27 ± 0.89	5.19 ± 0.57	25.86 ± 1.92 ^b^	2.54 ± 0.14	3.64 ± 0.33
mg Free AA/100 g PBD	0	16.20 ± 0.40	8.11 ± 0.76	2.94 ± 0.36	2.00 ± 0.30	18.16 ± 1.55	5.20 ± 0.80	9.66 ± 0.76
	175	21.12 ± 3.31	9.58 ± 1.15	3.01 ± 0.60	1.89 ± 0.13	38.13 ± 5.00	6.41 ± 1.63	10.96 ± 1.16
	364	13.43 ± 1.19	12.18 ± 0.93	2.77 ± 0.39	2.33 ± 0.26	44.01 ± 3.27	10.70 ± 0.60	9.51 ± 0.86

## Data Availability

The original contributions presented in the study are included in the article/Appendix A, further inquiries can be directed to the corresponding author.

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
