# Peer review of "Storage Stability of Plant-Based Drinks Related to Proteolysis and Generation of Free Amino Acids"

_foods, 2024, doi:10.3390/foods13030367_

Round 1

Reviewer 1 Report

Comments and Suggestions for Authors

The structure of the article is well-organized, with a clear introduction, body, and conclusion.

Here are my comments:

Line 57: You have chosen a storage period of one year. Why one year? Please explain. How is this one year in comparison to the shelf life?

Table 1: It would be interesting to know the shelf life indicated by the manufacturer, too. Maybe in “remaining months” from the start of the storage period?

Line 76: “Samples were then aliquoted, frozen at 80 °C and then transferred to -20 °C and stored until further analysis for other parameters.” So, was the particle size distribution determined from frozen samples? Freezing can influence particle size, particularly in the context of substances that contain liquids. Was this considered?

The pH measurement is not explained in the Materials and Methods section. Please provide some details (which pH meter etc.)

Figure 1 and Figure 2: It appears that the standard deviation is marked in the graphs. However, the labeling lacks the specification of the sample size n.

Author Response

Response to Reviewer’s comments

Authors: Thank you very much for your time on revising the paper and providing very constructive feedback. We appreciate your comments.

The structure of the article is well-organized, with a clear introduction, body, and conclusion.

Here are my comments:

Line 57: You have chosen a storage period of one year. Why one year? Please explain. How is this one year in comparison to the shelf life?

Authors: One year was chosen based on the shelf life of these commercial products, which were between 1-12 months for the selected PBD (as indicated in table 1).

Table 1: It would be interesting to know the shelf life indicated by the manufacturer, too. Maybe in “remaining months” from the start of the storage period?

Authors: Table 1 contains a row with the name “Shelf life” which indicates the shelf life according to the manufaturers and line 69 it states that all PBD were produced within 1-3 weeks prior to the experiment. Line 67 was updated for a clearer explanation of what info the table contains.

Line 76: “Samples were then aliquoted, frozen at 80 °C and then transferred to -20 °C and stored until further analysis for other parameters.” So, was the particle size distribution determined from frozen samples? Freezing can influence particle size, particularly in the context of substances that contain liquids. Was this considered?

Author: Yes particle size was measured on frozen samples, which we agree can potentially affect the size distribution. However, all samples were inspected visually and looked good and the evolving pattern for each PBD and small SD suggests that it has not been a main effect in our sample set.

The pH measurement is not explained in the Materials and Methods section. Please provide some details (which pH meter etc.)

Author: Description of pH measurement is now included in the text. Section 2.2.

Figure 1 and Figure 2: It appears that the standard deviation is marked in the graphs. However, the labeling lacks the specification of the sample size n.

Author: Sample size is now included in the figures and highlighted in the text.

Reviewer 2 Report

Comments and Suggestions for Authors

Dear Author,

I reviewed the manuscript:  Foods (ISSN 2304-8158)   entitled " Storage Stability of Plant-based Drinks Related to Proteolysis and Generation of Free Amino Acids ". 

This manuscript presents information about the stability of pH and particle size after a year-long storage period in commercial PBDs, along with colour development, proteolysis, and free AA measurement. Overall, the paper is within the scope of the journal; It is interesting, well organized and original. 

In my point of view, some issues should be reformulated and clarified to improve the manuscript:

-Legends in Table 2 and Figure 1 should be revised. It seems some problems occur because both legends contained comments to the results and that should not be included.

-Statistics results to some results are not evidenced in the tables/pictures. For example: in Table 2 in the pH and colour values; and Table 3 in all the PBDs except in Pee beverage. Could the authors include that data?

-The conclusion section should be revised. As the authors know, in this section the authors should summarize the key findings of their study, discuss their implications and if possible, give some practical recommendations.

Author Response

Response to reviewer’s comments

Dear Author,

I reviewed the manuscript:  Foods (ISSN 2304-8158)   entitled " Storage Stability of Plant-based Drinks Related to Proteolysis and Generation of Free Amino Acids ". 

This manuscript presents information about the stability of pH and particle size after a year-long storage period in commercial PBDs, along with colour development, proteolysis, and free AA measurement. Overall, the paper is within the scope of the journal; It is interesting, well organized and original. 

Authors: Thank you very much for your time on revising the paper and providing very constructive feedback. We appreciate your comments.

In my point of view, some issues should be reformulated and clarified to improve the manuscript:

-Legends in Table 2 and Figure 1 should be revised. It seems some problems occur because both legends contained comments to the results and that should not be included.

Author: The legends for Table 2 and Figure 1 have been revised accordingly.

-Statistics results to some results are not evidenced in the tables/pictures. For example: in Table 2 in the pH and colour values; and Table 3 in all the PBDs except in Pee beverage. Could the authors include that data?

Author: In Table 3, the letters indicating significant changes are included for those PBD and free AA where a significant difference was found. For non-significant results, we decided not to include the letters. The table legend was updated for clarity, line 243.

 -The conclusion section should be revised. As the authors know, in this section the authors should summarize the key findings of their study, discuss their implications and if possible, give some practical recommendations.

Author: The conclusion has been rewritten to follow the recommendation from the reviewer.

Reviewer 3 Report

Comments and Suggestions for Authors

The manuscript discusses changes in the properties of plant-based drinks (PBDs) during storage for 1 year. The study was carried out on 7 different commercial PBDs from the Danish market. Color, pH, particle size distribution, protein solubility, level of proteolysis and free amino acid generation were analyzed.

The topic and research plan of the study are interesting. However, in reviewer’s opinion, the manuscript needs significant improvements as noted below.

General comments

In the introduction, the authors outlined the idea and purpose of the study, but in my opinion, this goal was not achieved and presented in the manuscript satisfactorily. Please carefully analyze the entire manuscript and try to improve it, paying attention to the following issues:

- chaos throughout the text instead of a logical sequence of theses,

- the current form of the manuscript is only a simple report of the results obtained without any major logical connections prevails,

- captions of tables and figures should contain a concise, specific description of what they represent and an additional explanation of the symbols; however, they should not include a discussion of the results,

- little discussion explaining the phenomena, differences and similarities between individual PBDs,

- lack of connection between results and practice,

- few specific conclusions,

- the useful value of the results was not clearly presented,

- few comparisons of results with studies by other authors,

- sparse list of references used.

​

Detailed comments

-        line 11 – '(PBDs)' should be inserted because in lines 14, 15 the abbreviation PBD was used without explanation,

-        subsections 2.1, 3.1 – ‘colour’ or ‘color’ – please unify,

-        subsection 3.1 - changes in color parameters a* and b* during storage were not discussed,

-        line 189 - the name of the equipment for determining the particle size distribution has already been given in the 'Material and methods' section,

-        figure 1 and 2 -  I would suggest using the same colors for individual PBDs in both figures, and please consider using different point markers for individual PBDs to improve readability (some of the colors used are similar which makes reading difficult, especially the printed version),

-        lines 231-232 - please check 'L-alanine' - double described,

-        line 233 – please check ‘aminobutyric acid’ - it is not included in table 3,

-        line 236 - please check ‘ethanolamine’ - it is not included in table 3,

-        table 3 - please consider alphabetizing the amino acids for easier reading,

-        line 270 – the level of free AA in oat PBD increases significantly only in the range of 0 - 175 days,

-        lines 313-314 - missing commas.

Author Response

Response to reviewer’s comments

The manuscript discusses changes in the properties of plant-based drinks (PBDs) during storage for 1 year. The study was carried out on 7 different commercial PBDs from the Danish market. Color, pH, particle size distribution, protein solubility, level of proteolysis and free amino acid generation were analyzed.

The topic and research plan of the study are interesting. However, in reviewer’s opinion, the manuscript needs significant improvements as noted below.

Authors: Thank you very much for your time on revising the paper and coming with very constructive feedback. We appreciate your comments.

General comments

In the introduction, the authors outlined the idea and purpose of the study, but in my opinion, this goal was not achieved and presented in the manuscript satisfactorily. Please carefully analyze the entire manuscript and try to improve it, paying attention to the following issues:

- chaos throughout the text instead of a logical sequence of theses,

Author: The introduction has been rearranged and some parts rewritten for a better coherence throughout the text. Changes are highlighted.

- the current form of the manuscript is only a simple report of the results obtained without any major logical connections prevails,

Author: Major parts of the manuscript has been rewritten with this comment in mind. The introduction more clearly states the possible underlying mechanisms and these are followed into the discussion and conclusion.

- captions of tables and figures should contain a concise, specific description of what they represent and an additional explanation of the symbols; however, they should not include a discussion of the results,

Author: Legends to tables and figures have been changed to only contain relevant explanations to the figure/table and not the results.

- little discussion explaining the phenomena, differences and similarities between individual PBDs,

Author: The discussion has been revised to link the results and it is more clearly stated that these results need further investigations to follow up on what drives differences in PBD especially also in relation to plant source.

- lack of connection between results and practice,

Author: The discussion and conclusion has been rewritten. We do not observe any gelation in the products, which would be evident for the consumers but instead document changes that may affect the nutritional value or digestibility of PBD. Thereby this study raises important indications of quality deteriorations that should be explored further.

- few specific conclusions,

Author: The conclusion has been rewritten to follow the recommendation from the reviewer.

- the useful value of the results was not clearly presented,

Author: Changes conducted to improve the presentation of the results and the discussion of these.

- few comparisons of results with studies by other authors,

Author: More discussion has been added with comparison to other studies. Changes are highlighted in the text.

- sparse list of references used.

Author: More references have been included. ​

Detailed comments

-        line 11 – '(PBDs)' should be inserted because in lines 14, 15 the abbreviation PBD was used without explanation,

Author: The manuscript has been updated to include the correction mentioned above.

-        subsections 2.1, 3.1 – ‘colour’ or ‘color’ – please unify,

Author: The manuscript has been updated and unified to only using “color” as the way of spelling.

-        subsection 3.1 - changes in color parameters a* and b* during storage were not discussed,

Author: Section 3.1 has been updated to include parameter a* and b*.

-        line 189 - the name of the equipment for determining the particle size distribution has already been given in the 'Material and methods' section,

Author: The equiptment name was removed from line 189.

-        figure 1 and 2 -  I would suggest using the same colors for individual PBDs in both figures, and please consider using different point markers for individual PBDs to improve readability (some of the colors used are similar which makes reading difficult, especially the printed version),

-        lines 231-232 - please check 'L-alanine' - double described, line 233 – please check ‘aminobutyric acid’ - it is not included in table 3, line 236 - please check ‘ethanolamine’ - it is not included in table 3,

Author: This section was checked and changes were made according to the reviewers comments and highlighted in the text.

-        table 3 - please consider alphabetizing the amino acids for easier reading,

Author: The amino acids in Table 3 are alphabetised in such way that the essential AA are mentioned firstly in alphabetic order and then the non-essential AA in alphabetic order.

-        line 270 – the level of free AA in oat PBD increases significantly only in the range of 0 - 175 days,

Author: The text was updated to specify this point.

-        lines 313-314 - missing commas.

Author: Commas have been added to these lines.

Round 2

Reviewer 3 Report

Comments and Suggestions for Authors

The authors responded to most of my comments and revised the manuscript accordingly. In conclusion, I recommend the manuscript for publication in present form.

I only suggest correcting the title of Table 2 as follows:

Table 2. pH, color parameters (L*, a*, b*) and protein solubility measurement of stored PBDs on sampling days 0, 175 and 364. For pH and color parameters, the standard deviations were all under 3% and not included for better readability.’